# The Quality, Readability, Completeness, and Accuracy of PTSD Websites for Firefighters

**DOI:** 10.3390/ijerph17207629

**Published:** 2020-10-19

**Authors:** Shannon C. Killip, Natalie K. R. Kwong, Joy C. MacDermid, Amber J. Fletcher, Nicholas R. Carleton

**Affiliations:** 1School of Rehabilitation Science, McMaster University, Hamilton, ON L8S 4L8, Canada; jmacderm@uwo.ca; 2Department of Interdisciplinary Science, McMaster University, Hamilton, ON L8S 4L8, Canada; kwongn1@mcmaster.ca; 3Department of Physical Therapy and Surgery, Western University, London, ON N6A 3K7, Canada; 4Clinical Research Lab, Hand and Upper Limb Centre, St. Joseph’s Health Centre, London, ON N6A 4L6, Canada; 5Department of Sociology and Social Studies, University of Regina, Regina, SK S4S 0A2, Canada; amber.fletcher@uregina.ca; 6Anxiety and Illness Behaviours Laboratory, Department of Psychology, University of Regina, Regina, SK S4S 0A2, Canada; nick.carleton@uregina.ca

**Keywords:** posttraumatic stress disorder, operational stress injury, mental health, firefighters, first responders, public safety personnel, health resources, internet, website, readability

## Abstract

Firefighters appear at an increased risk for post-traumatic stress disorder (PTSD). Because of PTSD-related stigma, firefighters may search for information online. The current study evaluated the quality, readability, and completeness of PTSD online resources, and to determine how the online treatment recommendations align with current evidence. Google.ca (Canada) searches were performed using four phrases: ‘firefighter PTSD’, ‘firefighter operational stress’, ‘PTSD symptoms’, and ‘PTSD treatment’. The 75 websites identified were assessed using quality criteria for consumer health information (DISCERN), readability and health literacy statistics, content analysis, and a comparison of treatments mentioned to the current best evidence. The average DISCERN score was 43.8 out of 75 (indicating ‘fair’ quality), with 9 ‘poor’ websites (16–30), 31 ‘fair’ websites (31–45), 26 “good” websites (46–60), and nine excellent websites (61–75). The average grade level required to understand the health-related content was 10.6. The most mentioned content was PTSD symptoms (48/75 websites) and PTSD treatments (60/75 websites). The most frequently mentioned treatments were medications (41/75 websites) and cognitive behavioural therapy (40/75 websites). Cognitive behavioural therapy is supported by strong evidence, but evidence for medications appears inconsistent in current systematic reviews. Online PTSD resources exist for firefighters, but the information is challenging to read and lacks evidence-based treatment recommendations.

## 1. Introduction

Posttraumatic stress disorder (PTSD) is defined as a disorder resulting from exposure to one or more potentially psychologically traumatic events (PPTE) [1]. Events that can cause PTSD include life-threatening experiences or severe injuries due to violence, which causes feelings of distress and helplessness [2]. Not all individuals that experience a traumatic event will develop PTSD as it is dependent on coping strategies, supports, and past experiences [2,3]. The main symptoms of PTSD include recurrent memories or dreams about the event, avoiding negative reminders of the event, negative cognitions and mood, and marked increases in arousal and reactivity [3].

Firefighters and other public safety personnel are repeatedly exposed to diverse PPTE [4], making them more likely to develop mental and physical health injuries, including PTSD [4,5]. During the course of their careers, firefighters respond to hundreds of PPTE [4], including medical emergencies and violence [6]. In a study by Carleton et al. [7], 13.5% of Canadian firefighters screened positive for PTSD as per the Post-Traumatic Stress Disorder Checklist for DSM-5 (PCL-5). Recently, there has been increased interest in understanding the potential risks for firefighters developing PTSD and other operational stress injuries (OSI) [1], as well as the specific needs of firefighters for information and treatment [8]. There are barriers to firefighters receiving treatments. The firefighting community [9] and society in general [10] experience strong stigma regarding mental health challenges, which is a barrier to seeking treatment. Due to the ‘macho mentality’ prevalent in the male-dominated occupation of firefighting, firefighters may be seen as weak if they report mental health issues [8]. Firefighters may be worried about their reputation and see mental illness as a threat to their career, thus they may be less likely to report PTSD or seek professional help [8]. The main reasons firefighters in South Korea do not seek treatment for PTSD involve concerns about potential stigma and a perceived lack of availability of treatment [8]. First responders collectively appear less likely to seek treatment for PTSD than the general population due to the associated stigma, the potential loss of privacy, and a culture of self-reliance [5]. Firefighters might be more inclined to look online for information about PTSD symptoms and treatments due to the convenience, accessibility, and anonymity of the Internet, which can help them to avoid stigma [11,12]. Firefighters may also use online health resources because of insufficient resourcing by their fire department or their community. However, many local fire department websites lack online mental, physical, and general health resources [13]. The Internet is an important resource for firefighters and other public safety personnel who are seeking information and potential treatment for work-related PTSD and other OSIs.

The Internet is an increasingly popular tool for finding health information and is often one of the first steps for seeking social support [11], professional help [14], and treatments [15]. The most recent published Canadian Internet Use Survey collected in 2012 found that 67% of Canadians used the internet to find health-related information [16]. Similarly, a 2012 survey evidenced most Americans (59%) sought health information online in the past year using search engines such as Google, Bing, or Yahoo, and many (35%) used the Internet to diagnose a medical condition; however, only half (53%) discussed their online findings with health care professionals [17]. There are important concerns regarding the quality, readability, completeness, and accuracy of the health information on the Internet [11]. As anyone can publish information online and the trustworthiness of a website can be difficult to discern [11,18,19]. Consumers of health information online are unlikely to be medical professionals and cannot always appropriately evaluate for the quality and accuracy of online information [20], and cannot always understand the complex medical content provided online due to insufficient literacy or inadequate health literacy skills [21]. Unfortunately, websites of better quality and content tend to be more difficult to read and comprehend [22]. Accuracy, relevance, and accessibility of information are all important facets of efficient use of health information. Past studies assessing online health information for different health populations have found that many websites lack accuracy and completeness of the content and are difficult to comprehend for laypersons [23,24]. Other studies assessing the quality of health information websites have found that many make unsupported claims, contain bias, and lack details such as the risks associated with treatments [25,26,27]. The quality concerns mean people can be misled with confusing or incorrect health information that can compromise healthcare decisions making [27,28].

There are general population studies assessing the quality of online health information; however, none specifically review online resources for PTSD in firefighters. Firefighters tend to prefer firefighter-specific resources for mental health help and suicide prevention, such as online resources, rather than general resources such as help hotlines [9]. PTSD resources that are specific to firefighter needs and their occupational demands are more likely to be accepted by, usable for, and useful to firefighters. Firefighter-specific PTSD resources exist [13], but the quality, readability, and accuracy of the included content remain unknown. There is also no evidence that appropriate PTSD resources can be easily found using a simple Internet search rather than depending on firefighter association websites [13].

Research is needed to identify and assess the quality of PTSD resources generated from Internet searches that would be performed by firefighters. The current study was designed to conduct a search of websites to identify online resources for firefighters with PTSD and to evaluate the quality, readability, and completeness of the content of these websites. The current study was also designed to determine how closely the website treatment recommendations align with current best evidence in order to gain a thorough understanding of what resources are currently available online for firefighters with PTSD.

## 2. Materials and Methods

The current study used a review of current website resources for PTSD. Keyword searches were performed, and websites were selected based on relevance. A standardized quality tool and readability statistics were used to appraise the selected websites. The current methods have been used previously in a review of websites for fibromyalgia [22]. Relevant content was extracted from the websites, and the accuracy of the content was also assessed.

### 2.1. Search Strategies for Identifying the Online Resources

There were four keyword searches used in the current study. There were four full-time firefighters from four Canadian provinces (Hamilton, Ontario; Montreal, Quebec; Vancouver, British Columbia; and St. John’s, Newfoundland) who were asked what keywords they might use to search for PTSD information. Commonly mentioned search terms included firefighter-specific phrases (i.e., ‘firefighter PTSD’, ‘firefighter operational stress’), and general PTSD search terms (i.e., ‘PTSD symptoms’, ‘PTSD treatment’). The keyword searches were performed from a Canadian location using Google.ca (Google Canada) (Google Inc., California, USA) in July 2019. Google.ca was used as the search was performed in Canada. An ‘incognito browser’ was used to avoid search bias based on the internet search history and to ensure that the search term results did not influence subsequent search term results. Most people searching for information online will stop at the second page of Google results, meaning approximately 20 results are typically reviewed [29]. The first 20 results that fit the inclusion criteria for each keyword search were included in this study unless 20 relevant results could not be found. If less than 20 results were found for a search term, all relevant results were included. The name of the websites and the website hyperlinks were recorded in a spreadsheet organized by search term.

### 2.2. Criteria for Selecting Online Resources

Several criteria were set for the inclusion or exclusion of a website. Included websites needed to present information in English about PTSD or OSIs. Duplicates were removed, meaning that if a resource appeared in multiple searches, it was only included once. Social media websites were excluded as this study focused on formal online resources; however, online peer support resources as well as social media pages are becoming increasingly important in mental health care [30].

### 2.3. Quality Appraisal and Readability Tools

DISCERN (quality criteria for consumer health information on treatment choices) is a tool developed by an expert panel for evaluating the publication reliability and quality of written health information [28]. The DISCERN score of a website has been correlated with overall website quality [31,32]. The DISCERN score cannot be used to determine the accuracy or completeness of the content but was developed for assisting individuals without prior topic-specific knowledge to assess the quality and potential biases of the information [28]. DISCERN has acceptable inter-rater reliability when assessing websites on social phobia [33] and pediatric neuro-oncology [33]. DISCERN has good face validity and content validity relative to other similar measures [33,34], appears to have broad applicability [28], and has previously been used to evaluate online resources for specific health conditions [22,33,34].

All identified websites were manually rated by a member of the research team based on the DISCERN guidelines [35,36]. The websites were rated on a scale from 1 to 5 across the 15 criteria. An overall rating of the website was also given on a scale of 1 to 5. A score of 1 indicates that the text did not fulfill the criterion. A score from 2 to 4 indicates that the text fulfills the criterion to some extent. A score of 5 indicates that the text fulfills the criterion completely. An overall DISCERN score was calculated out of 75, which was the maximum overall score for the 15 categories of the DISCERN instrument which score specific components of a text [22,35,36]. Accordingly, websites could be assessed as a whole using DISCERN and categorized into excellent (61–75), good (46–60), fair (31–45), poor (16–30), or very poor (<16) [22]. One rater evaluated the website quality using DISCERN for all the websites included in the current study. The first three randomly selected websites were evaluated by the primary rater and a second rater using the same DISCERN criteria to check for rating bias and to make sure the primary rater understood how to use the rating system. After performing the quality assessment for the three websites separately, the two evaluators met to discuss their ratings until a consensus was reached. There was agreement between the raters for 2 of the websites. One website was found to have different ratings as the first rater had reversed the rating scale. This error was discussed and corrected.

There were three readability assessments conducted on the text of each website. The Flesch–Kincaid Grade Level formula estimates the grade level required to read a certain piece of text. The Flesch reading ease (FRE) score also gave an indicator of the readability of the text. The FRE is scored on a scale from 0 to 100 [37,38]. Texts with higher scores have a higher ease of reading, with a score of 90–100 being very easy, 80–89 being easy, 70–79 being fairly easy, 60–69 being standard, 50–59 being fairly difficult, 30–49 being difficult, and 0–29 being very confusing [22]. A health literacy tool called the Simple Measure of Gobbledygook (SMOG) was used to assess the required education level to understand health-related text [39]. Health information should be written at a sixth–to eighth grade readability level to ensure good readability and comprehension for most adults; as such, a Flesch–Kincaid grade level of 6–8, an FRE score of approximately 60 or higher [37,40], and a SMOG approximate grade level of 8 would be appropriate. Lower readability grade levels improve the accessibility for firefighters and the general public [37,40], which may be critical for persons seeking resources when fatigued or experiencing a mental health injury.

### 2.4. Data Extraction for Content Analysis

Content analyses were performed to assess the completeness of the websites. An original list of codes specific to PTSD online content for firefighters was developed a priori to aid in the extraction of data from the websites. The codes were generated by two co-authors with research experience in the field of mental health and PTSD in public safety personnel, and then these codes were revised by the entire research team. The final list of codes was developed by co-author consensus. The codes were used to label the descriptive data collected in order to organize and summarize the information. The codes and the guiding definitions used in the data extraction included: recommended PTSD treatment and program details (e.g., type of program or treatment, delivery methods, time to completion, associated costs, evidence of effectiveness, associated pseudoscience), treatment providers (e.g., clinical psychologist, certified trainer, peer), website’s target audience (e.g., does the website specifically mention a target audience population such as firefighters, or is the target audience more general such as people with PTSD looking for support programs and general information seekers), self-help advice provided, and information about PTSD (e.g., causes, diagnosis, symptoms, quality of life). Categorized information was summarized in a data extraction table. If content for a certain code was not found on a website, the section was marked as ‘not applicable’ (Appendix A: Content Analysis Codebook).

### 2.5. The Development of Evidence Based Statements

DISCERN does not assess the accuracy of the information provided in the websites [34]; therefore, evidence-based statements were created in order to evaluate whether the content of the website corresponded to the best-published evidence in the literature [11]. There were 10 evidence-based statements developed after reviewing literature about the effectiveness of PTSD treatments. Evidence from systematic reviews, randomized control studies, and one review report specific to firefighters and PTSD were selected as references for the statements. For treatments with either strong evidence or no evidence, the websites were categorized as agreeing with the evidence-based statement, disagreeing with the evidence-based statement, or not mentioning the treatment. A slightly different system was used for statements regarding content with inconclusive evidence as the website could disagree with the evidence in two ways: the website might only report evidence supporting the treatment (i.e., categorizing the website recommendation as ‘disagrees with a focus on positive evidence’), or the website might only report evidence that does not support the treatment (i.e., categorizing the website recommendation as ‘disagrees with a focus on negative evidence’). If the website stated that the evidence was inconclusive (i.e., the website provides both supporting and opposing evidence for the treatment) or did not mention a treatment at all, ‘agree’ and ‘not mentioned’ were also potential categories, respectively. Using evidence-based statements supported evaluating the accuracy of website information, which the DISCERN does not assess [29]. When the information extracted from the online resources is consistent with the best evidence found in the literature, online resources can then be considered to contain high-quality information [41,42].

One researcher rated the treatment recommendations for each website. For each treatment included in the 10 evidence-based statements, the researcher recorded whether or not the treatment was mentioned. The treatment text was then compared to the corresponding evidence-based statement and the most appropriate response was selected. Any recommended treatments that were mentioned on the website but not identified as evidence-based in the published systematic reviews for PTSD treatments were recorded for interest, but the associated text was not evaluated for accuracy because of the absence of sufficient evidence.

### 2.6. Website Ranking

The websites were ranked by three different criteria: (1) total DISCERN score; (2) Flesch–Kincaid Grade level; and (3) the number of recommended treatments corresponding with the evidence-based statements. Results from all four search terms were tabulated in a spreadsheet and used to find the 20 websites with the highest values for each category. The websites that mentioned treatments supported by strong evidence were ranked higher on evidence-based statements.

### 2.7. Data Analysis

The data collected from the DISCERN, readability tests, content analysis, and the evidence-based statements were organized into tables in Microsoft Excel (Microsoft, Redmond, USA). Simple summary statistics were used to present the data. The calculations of averages and frequencies were performed in Microsoft Excel.

## 3. Results

After excluding 16 websites (6 duplicates, 3 new reports, 6 had insufficient information or were incomplete, 1 redirect page leading to a website included from a previous search), the first 20 websites that met the inclusion and exclusion criteria were selected from each search term. All search terms generated 20 relevant websites, except the search term ‘firefighter operational stress’ which only generated 15 relevant results. The search process identified 75 websites.

Of the 75 websites, 19 were firefighter specific. The search term ‘PTSD Treatment’ did not generate any websites specific to firefighters and only generated one website specific to first responder specific. The other identified websites were designed for the general population (14 of the 20 websites), veterans (4 of the 20 websites), and psychologists (1 of the 20 websites). The search term ‘PTSD Symptoms’ only generated one firefighter specific website. The other websites were designed for the general population (18 of the 20 websites), and veterans (1 of the 20 websites). The search term ‘Firefighter PTSD’ generated 18 websites specific to firefighters, one website specific to first responders, and one general information website. Among the 18 websites specific to firefighters, one website also contained information for the general public, and two websites had content designed for researchers working with PTSD in firefighters. The search term ‘Firefighter Operational Stress’ was more specific to military personnel and veterans. All 15 websites were designed, at least in part, for veterans, and most also contained content for current Canadian Forces personnel, members of the Royal Canadian Mountain Police (RCMP), and retired RCMP. Among the 15 websites, three mentioned first responders, but none specifically mentioned firefighters.

The average DISCERN score for all 75 websites was 43.8 (indicating ‘fair’ quality on average). The results had varying average DISCERN scores by keyword search: specifically, 38.6 for firefighter PTSD, 43.6 for firefighter operational stress, 42.6 for PTSD symptoms, 51.6 for PTSD treatment (Table 1). Ranking the websites based on the overall DISCERN rankings produced the following results: nine websites (12%) were ranked as ‘poor’ (score range 16–30), 31 (41%) were ranked as ‘fair’ (score range 31–45), 26 (34%) were ranked as ‘good’ (score range 46–60), and nine (12%) were ranked as ‘excellent’ (score range 61–75). There were no websites classified as ‘very poor’ (<16). The number of websites that addressed (DISCERN score of 5), partially addressed (DISCERN score of 2–4), and did not address (DISCERN score of 1) each of the DISCERN criteria is presented in Figure 1. The DISCERN criteria that were most successfully addressed among the 75 websites included criterion 1 (clarity of the website’s aims) with an average score of 4, criterion 2 (did the website achieve the intended aims) with an average score of 4.3, and criterion 3 (is the website relevant to the user) with an average score of 4.1 (Figure 2). The DISCERN criteria that were the most poorly addressed among the 75 websites included criterion 4 (clear referencing of the sources of information) with an average score of 1.9, criterion 5 (does the website provide publication dates for the sources of the information) with an average score of 2.3, criterion 10 (are the benefits of each treatment explained) with an average score of 2.4, criterion 11 (are the risks of each treatment explained) with an average score of 1.9, and criterion 12 (does the website explain what happens when the condition is not treated) with an average score of 1.9 (Figure 2).

The average FRE score was 51.7, indicating readability for the 75 included websites was ‘fairly difficult’. The average Flesch–Kincaid grade level required to understand the written material was 8.9, and the health literacy SMOG grade level to understand health-related written material was 10.6. The readability scores varied across the keyword searches. The keyword search for ‘PTSD treatment’ produced the highest average FRE score (i.e., 54.4), indicating content that was ‘fairly difficult’ to read [22]. The keyword search for ‘firefighter operational stress’ produced the lowest average FRE score (46.2), indicating content that was ‘difficult to read’ based on the rating scale [22]. The average readability statistics for each keyword search are summarized in Table 1 (Appendix A: DISCERN and readability scores).

The number of websites per search term that mentioned content relevant to the codes used for the content analysis are presented in Table 2. The most commonly identified content throughout all 75 websites was related to the code ‘information about PTSD’ which described general PTSD information other than treatment details. Only 5 of the 75 websites (6.7%) did not include some form of information about PTSD. Many of the websites also mentioned some form of treatment. The keyword search ‘PTSD treatment’ returned the highest percentage of results that mentioned treatment, with all 20 websites (100%) mentioning at least one treatment. Less frequently mentioned in the 75 websites were self-help advice (59%) and PTSD programs (39%). Almost all websites (14/15) generated from the search term ‘firefighter operational stress’ were related to PTSD treatment programs that mainly focused on veterans, military personnel, and RCMP members (Appendix A: Coding sheet).

There were ten evidence-based statements regarding PTSD treatment effectiveness based on results from randomized control studies and systematic reviews. Strong evidence was found supporting the effectiveness of exposure therapies [43,44,45], cognitive-behavioural therapy (CBT) [43,44,45,46,47], and eye movement desensitisation and reprocessing (EMDR) [45,46,47,48] for treating PTSD. Inconclusive evidence was found for the effectiveness of mindfulness-based stress reduction and meditation-based treatments [43,49], stress management interventions [45], non-trauma focused interventions (i.e., supportive therapy, non-directive counselling, psychodynamic therapy and hypnotherapy) [45,46], virtual reality exposure therapy [44,50], medications (i.e., sertraline, paroxetine or others) [44,51,52], and crisis-focused psychological interventions [53] for treating PTSD. No evidence was found supporting the effectiveness of peer support programs for the prevention and treatment of PTSD symptoms in firefighters [53].

The treatment information found on the 75 websites based on the evidence-based statements is summarized in Figure 3 and Figure 4. No single website mentioned all of the treatments that were highlighted in the evidence-based statements. The most frequently mentioned treatments across the 75 websites were cognitive behavioural therapy (CBT) (40 mentions), non-trauma focused therapies (30 mentions), and medications (41 mentions), of which only CBT is supported by strong evidence. The effectiveness evidence for non-trauma focused therapies and medications is inconclusive; nevertheless, most websites only mentioned evidence supporting the benefits of such treatments for PTSD. Only one website that mentioned non-trauma focused therapies and one website that mentioned medications consistent with the evidence-based statements for these treatments. The least frequently mentioned treatments in the 75 websites were crisis-focused treatments (2 mentions), virtual reality exposure therapy (4 mentions), and stress management techniques (5 mentions). Some websites included other potential treatments which were not included in the evidence-based statements due to insufficient evidence from randomized control studies and systematic reviews. These treatments included float therapy (mentioned by 1 website), support dogs (mentioned by 1 website), yoga (mentioned by 2 websites), methylenedioxy-methylamphetamine (MDMA) therapy (mentioned by 1 website), equine therapy (mentioned by 1 website), and essential oils (mentioned by 1 website). The potential treatments were noted for interest but excluded from the evidence-based statement statistics. A total of 5 websites mentioned treatments that were not included in the evidence-based statements. One website also mentioned inpatient therapy, but the website did not provide any other details as to what types of treatments could be used during inpatient therapy (Appendix A: Evidence-based statements).

The websites with the top 10 DISCCERN scores, readability scores, and the accuracy scores for PTSD treatment recommendations are presented in Table 3. Many websites excelled in one domain but performed poorly in others. For example, the American Psychological Association’s website cited many studies highlighting both the benefits and the drawbacks to almost all treatments and had an excellent DISCERN score of 68; however, the same website had very poor readability and health literacy statistics (FRE of 27.2, Flesch–Kincaid Grade Level of 11.4, SMOG grade of 12.3), and therefore excluded from the top 10 PTSD websites.

## 4. Discussion

Results from the current study suggest that websites designed to provide PTSD information to the general public or specifically for firefighters have varying quality, health literacy, and readability levels, and that their content often does not align with the current best evidence for treatment recommendations.

The average DISCERN rating for all 75 websites was ‘fair’, with individual website DISCERN ratings ranging from ‘poor’ to ‘excellent’. The current study found that, on average, online PTSD resources were of fair quality except for some important concerns. The DISCERN results showed that many websites did not reference the sources of their information. There is a large amount of online health information, but very little is evidence-based [28]; concordantly, there is no way to tell whether the information presented on PTSD websites comes from reputable and unbiased sources. A study by Griffiths and Christensen [31] on the quality of online depression health resources found similar results as only 11 of the 21 websites included in their study included referencing. The authors also explain that the referencing in the 11 websites was not performed consistently throughout the entire website.

Many PTSD websites in the current study did not adequately discuss the risks and benefits related to the presented treatments mentioned. The results are consistent with research indicating online health resources tend not to address the risks and benefits of treatments [22,28], focusing on the treatment processes rather than the outcomes [28]. For example, one study that also used DISCERN to assess the quality of fibromyalgia websites and found that half of the 25 websites addressed or partially addressed the benefits of the treatments listed [22]. This study also found that only one website fully addressed the risks and four websites partially addressed the risks of the treatments listed. Information about the risks and benefits of treatment is important for making informed treatment decisions [28,64]. Unfortunately, when websites do not provide information about the treatment risks and benefits, people may struggle to select the treatment best suited for their needs [28].

Websites in the current study that were of higher quality were often associated with poor readability. For example, the website Everyone Goes Home provides a PTSD resource specific to firefighters with a good quality website (DISCERN rating of 59), but difficult to read (i.e., a Flesch Reading Ease of 35.7, a Flesch–Kincaid grade of 13.2, a SMOG grade of 11.9). The pattern of results accords with previous research on fibromyalgia websites as the study found that the websites with the best DISCERN scores also had high readability requirements rated [22]. Conversely, in the current study, some websites were found to be of good quality with acceptable readability levels, as seen in the top 10 websites (Table 3).

The current study identified that there is insufficient high quality and comprehensible information online that satisfies the information needs of firefighters wanting to learn more about PTSD. Many websites in the current study provided in-depth treatment details, but the health literacy requirements were so high as the average grade level to understand the health content for all 75 was 10.6. These websites may be confusing, misleading, or inaccessible to people without health-related expertise. The Canadian Council on Learning estimates that at least 60% of Canadians have low levels of health literacy, meaning that low health literacy is sufficiently prevalent that hard-to-read websites will be a barrier for many people [65]. Health information may be particularly difficult to read and understand without a scientific background [66]. Accordingly, valued and trusted Associations should be providing website content at a reading grade level of 6-8 to facilitate most adults readily understanding and successfully using the information [37,40]. During the creation of PTSD and other health resources, as well as for health research in general, it is important to consider target audiences and use language that is accessible to the audience through strategies such as defining complex terms or creating lay summaries. Firefighters in Canada may have a wide range of health literacy levels. The firefighter population in Canada consists of both volunteer and career firefighters. Based on the NFPA Canadian Fire Department Profile, 17% of Canada’s firefighters are career firefighters, while 83% are volunteers [67]. Because career firefighters are often required to complete a relevant college-level program before they can work as firefighters [68] it is possible that they may have higher literacy levels compared to volunteer firefighters. Because firefighting is not a career for volunteer firefighters and they have a range of different jobs, volunteer firefighters can also have a wide range of education levels and thus, their readability and health literacy levels can fluctuate. For example, in a study of 100 volunteer firefighters in Prince Edward Island (PEI), Canada, 68% of volunteer firefighters explained that they never received education on mental health [68]. Because such a large range of literacy levels exist amongst firefighters in Canada, it can be difficult to target a specific audience for online resource creation. With this in mind, future online PTSD resources should be created specific to volunteer firefighters and specific to career firefighters. If this is not feasible, the resources should be accessible to all firefighters by using simple language, targeting a reading level of grade 6 to grade 8.

There was a large variety of content presented on the 75 websites. Most websites in the current study presented PTSD information such as risk factors, diagnosis, and symptoms. Only 5 websites did not present any PTSD information and instead, solely promoted PTSD programs and the treatments provided by the website owners. The most commonly presented PTSD information included signs and symptoms (49 of the 75 websites) and treatments for PTSD (60 of the 75 websites). The keyword search influenced what content was returned. All 20 websites generated from the ‘PTSD treatment’ search contained treatment information, but only 14 of the 20 websites from the ‘PTSD symptoms’ search contained treatment information. Many websites did not mention specific programs or services that people with PTSD could access for help, and the websites that did were almost exclusively for the Armed Forces members or veterans. Through the searches performed, there were few PTSD programs designed specifically for first responders or other public safety personnel. Websites that did offer support to firefighters often gave general information about treatment options and self-help. Advertised support programs often lacked detail, providing only contact information or help hotline phone numbers. Interestingly, all of the 15 websites identified from the search term ‘firefighter operational stress’ were targeted, at least in part, to veterans, even though the term operational stress is often used for both public safety personnel as well as members of the Armed Forces and veterans [59]. Although the term ‘firefighter’ was used in the search, none of the 15 websites mentioned being targeted for firefighters, and only 3 of the 15 websites mentioned first responders. Because of this, firefighters should consider using the search term ‘operational stress’ if they are looking for firefighter specific online resources. PTSD programs specifically for firefighters should be created, independently empirically evaluated, and made accessible online using both ‘PTSD’ and ‘operational stress’ as keywords. PTSD programs and resources exist through certain fire departments and firefighter associations [13], but firefighters may want to access PTSD supports online due to concerns about stigma or accessibility (e.g., geography and time can be barriers) [8].

Of the 60 websites that mentioned treatments, the most frequently mentioned treatments were medications, cognitive behavioural therapy, and non-trauma focused therapies, although medications and non-trauma focused therapies were not supported by conclusive evidence. Websites in the current study did mention treatments supported by strong evidence, but many promoted the use of treatments with inconclusive evidence or no evidence. When there were inconsistencies between the available evidence and the presented content, all but two of the websites included only the research results that supported the treatments and did not mention any evidence against the use of the treatment for PTSD. The presentation bias can lead firefighters to overlook treatments with strong evidence, selecting instead a treatment with inconclusive evidence. Similar results were found in a study by Griffiths and Christensen [31] who assessed the quality of online information for depression. The authors explained that many of the websites they reviewed did not mention scientific evidence to support their claims. In addition, the authors found websites that did include evidence to support the treatment recommendations did not elaborate on the strength of the evidence that they referenced [31]. Moreover, Griffiths and Christensen identified 53 different interventions recommended for depression across 21 websites, many of which were presented without scientific evidence [31]. Similarly, in the current study, some websites promoted treatments without evidence-based support from randomized control studies and systematic reviews (e.g., equine therapy, essential oils, or yoga), and some websites even stated that these treatments were ‘evidence-based’ without providing a reference. Firefighters need to carefully and critically assess with the information available online to support actual evidence-based decision-making regarding their health and avoid potentially ineffective and expensive treatments that can hinder their recovery.

### Limitations of This Study

The keyword searches were conducted with Google.ca. The results might have varied across different search engines. The results might have also been different if a different Google location was used (i.e., Google.com). Because the search was performed in Canada, Google.ca was used, and thus, Canadian websites were primarily identified. It is likely that if this search was performed in another country, that other websites with different content, target audiences, and health care practices would be found. Because of this, the results of the content analysis are most likely specific to a Canadian Google search, and thus most relevant to Canadian firefighters. Although both English and French are the official languages of Canada, only English websites were searched. It is a limitation that French websites and websites of other languages were not included in the study. Moreover, the keywords selected had a large impact on the websites that were found in the current study. For example, the term ‘operational stress’ is often used for public safety personnel (i.e., firefighters, paramedics, police officers, RCMP) as well as for military personnel and veterans (i.e., past or present members of the Canadian Armed Forces) [7], and thus this influenced the results of the content analysis. Only four keywords were used in the searches. Firefighter searching for information online might use more or different keywords and therefore find more information beyond what was assessed in the current study. For example, firefighters looking for PTSD support programs could specifically use the keywords ‘PTSD support program for firefighters’. The DISCERN tool has been used in other research, but results from a different quality assessment tool might vary. The DISCERN scoring tool also requires researchers to interpret the guidelines and the website content to rate the quality, which means the quality appraisal necessarily includes bias. Only the recommended treatments were assessed for accuracy and compared to evidence-based statements generated from the literature. The websites contained other content about PTSD, but the accuracy of this content was not evaluated. The overall quality and accuracy of the entire text of a website can be difficult to ascertain and the congruence between evidence and the statements on the website was only partially assessed. The state of evidence is evolving, which means congruence of evidence across websites may change over time.

## 5. Conclusions

The quality, readability levels, completeness of the content, and accuracy of the treatment recommendation of websites about PTSD are not sufficient for the needs of firefighters. The current results demonstrate that many online resources for firefighters with PTSD do not have adequate information. Websites with adequate information are often not accessible due to complicated language and low readability levels. The websites sampled in the current study often lacked comprehensive information about PTSD and where firefighters can go to get help. The websites that did provide information about PTSD and the associated treatments rarely provided evidence or reference for their claims. After comparing the website information about treatments to evidence-based statements created from a literature review, many PTSD online resources were found to promote treatments that have inconclusive or no evidence. Lastly, the current study identified that many PTSD resources found online were not specific to firefighters.

Future research should identify the most relevant PTSD information, treatments and resources specifically for firefighters. A website can then be created specifically for firefighters that can be easily accessed online through an Internet search. The website should be high-quality, should reference PTSD information, and should explain the benefits and risks of all suggested treatments. The website should be easy to read and understand, by targeting a readability level of grade 6 to grade 8 and by explaining all complex scientific ideas in simple lay terms.

## Figures and Tables

**Figure 1 ijerph-17-07629-f001:**
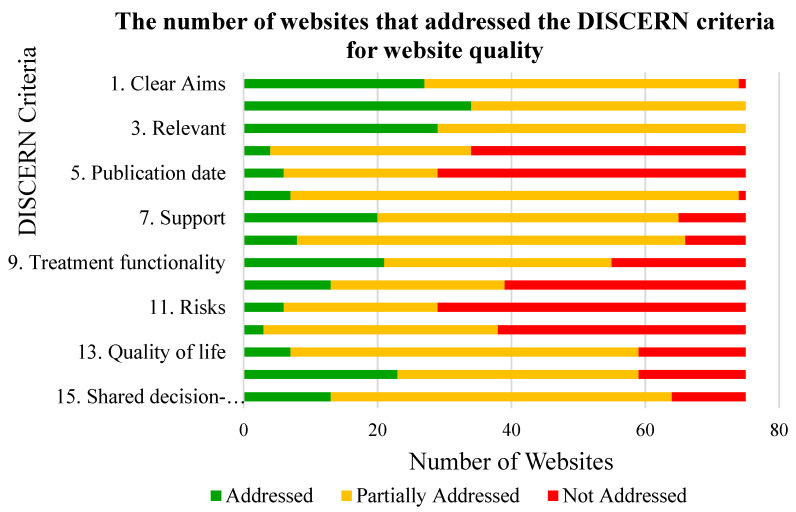
The number of websites out of the total 75 websites that addressed, partially addressed, and did not address each of the 15 DISCERN criteria. The DISCERN criteria include: (1) Are the aims of the website clear and is the target population of the website clear; (2) Does the website achieve the intended aims; (3) Is the website relevant to the needs of the user (in this case, firefighters with post-traumatic stress disorder (PTSD)); (4) Are the sources of information used for the website clear (i.e., proper referencing); (5) Does the website provide publication dates for the sources of the information included; (6) Does the website provide balanced and unbiased information (i.e., regarding treatment options for PTSD); (7) Does the website provide links to additional sources of information and support for treatment choices; (8) Does the website refer to area of uncertainty when it comes to the best treatment choice; (9) Does the website include information about how the treatments work (i.e., effects on the body, the condition, and the symptoms); (10) Are the benefits of each treatment explained; (11) Are the risks of each treatment explained; (12) Does the website explain what happens when the condition is not treated; (13) Does the website explain how the treatment will affect quality of life and activities of daily living; (14) Does the website make it clear that there are different possible treatment options; (15) Does the website promote discussing treatment choices with all involved in the patient’s care (i.e., shared decision-making) [35,36].

**Figure 2 ijerph-17-07629-f002:**
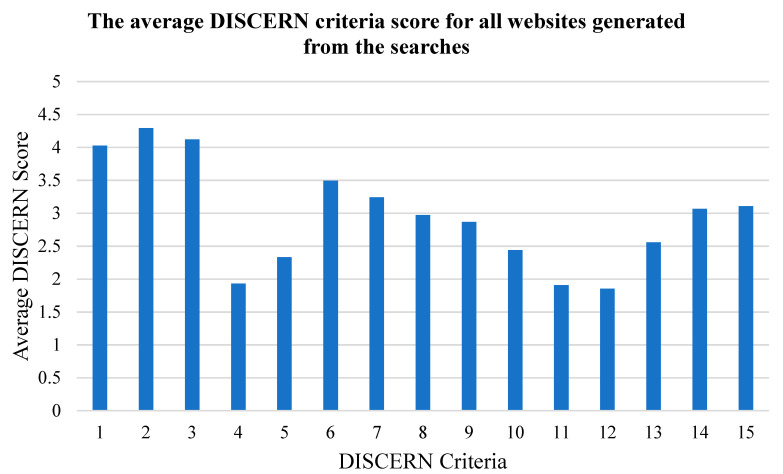
The average score of each of the 15 DISCERN criteria, based on the average of all 75 websites from the searches. Each DISCERN criteria is rated between 1 (does not address) and 5 (does address). The DISCERN criteria definitions can be found in Figure 1.

**Figure 3 ijerph-17-07629-f003:**
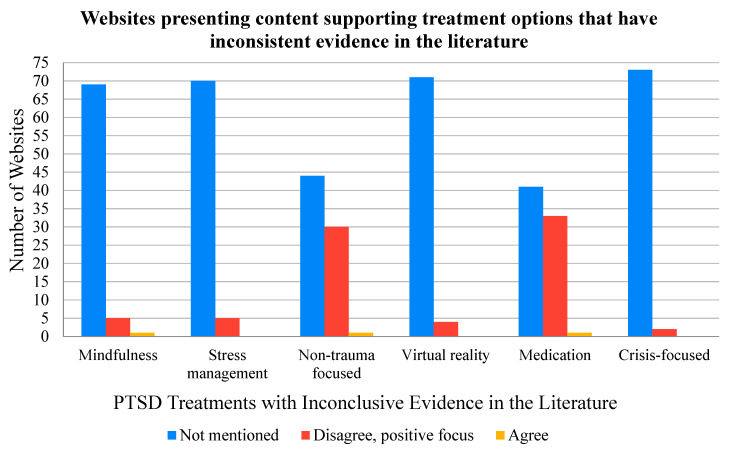
The number of websites that agreed or disagreed with the evidence-based statements for the potential treatments that have inconclusive evidence to support their effectiveness. Additionally included are the number of studies that did not mention the potential treatments. All studies that disagreed with evidence-statements had a positive focus and only presented the evidence to support the treatment effectiveness. None of the websites disagreed with a negative focus, meaning that the unsuccessful treatment of PTSD was not mentioned.

**Figure 4 ijerph-17-07629-f004:**
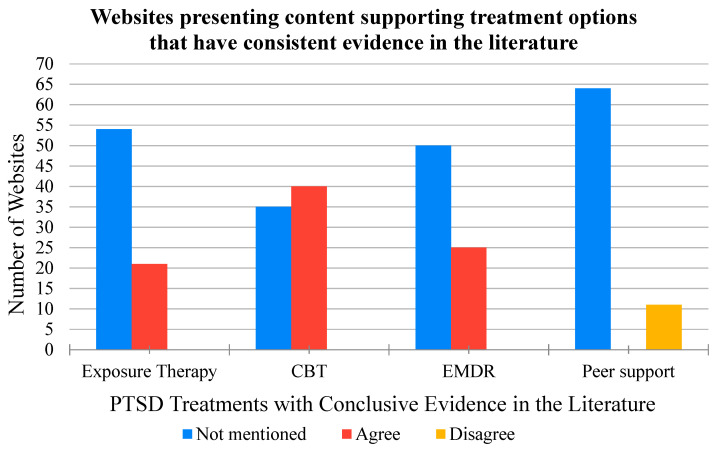
The number of websites that agreed or disagreed with the evidence-based statements that these established treatments have either strong evidence or no evidence to support their effectiveness. Additionally included are the number of websites that did not mention the treatments. For the treatments with strong evidence to support effective PTSD treatment (i.e., exposure therapy, cognitive-behavioural therapy (CBT), eye movement desensitisation and reprocessing (EMDR)), none of the websites disagreed with the evidence-based statements. For peer support, which has no evidence as an effective PTSD treatment, none of the websites agreed with the evidence-based statements.

**Table 1 ijerph-17-07629-t001:** The average readability statistics and DISCERN scores for the websites are generated from each of the keyword search terms. The average DISCERN total scores are out of a total possible score of 75, with higher scores representing better quality websites. The average Flesch reading ease (FRE) scores are out of a total possible score of 100, with higher scores being simpler to read. The average Flesch-Kincaid Grade and average Simple Measure of Gobbledygook (SMOG) levels represent school grade levels, with lower grades being simpler to read and understand.

Keyword Search Terms	Average DISCERN Total Score	Average FRE	Average Flesch-Kincaid Grade	Average SMOG
Firefighter PTSD (*n* = 20 websites)	38.6 (fair)	53.8 (fairly difficult)	8.8	11.6
Firefighter Operational Stress (*n* = 15 websites)	43.6 (fair)	46.2 (difficult)	9.1	11.0
PTSD Symptoms (*n* = 20 websites)	42.6 (fair)	51.0 (fairly difficult)	9.8	9.3
PTSD Treatment (*n* = 20 websites)	51.7 (good)	54.4 (fairly difficult)	7.8	10.5

**Table 2 ijerph-17-07629-t002:** The frequency of websites that mention the codes from the content analysis based on the different search terms. The number of websites that have mentioned the content is provided as a fraction of the total number of websites generated from the search term and the corresponding percentage.

Coding Categories		Keyword Search Terms			
		Firefighter PTSD (*n* = 20)	Firefighter Operational Stress (*n* = 15)	PTSD Symptoms (*n* = 20)	PTSD Treatment (*n* = 20)
Target Audience	General Audience (*n* = 35)	4/20 20%	2/15 13%	16/20 80%	12/20 60%
	Specific Audience (i.e., veteran, first responder) (*n* = 22)	16/20 80%	1/15 7%	2/20 10%	4/20 20%
	Individuals looking for PTSD programs (*n* = 18)	0/20 0%	12/15 80%	2/20 10%	4/20 20%
Websites that Mentioned Treatments (*n* = 60)		13/20 65%	13/15 87%	14/20 70%	20/20 100%
Treatment Provider	General Healthcare Provider (*n* = 21)	3/20 15%	8/15 53%	1/20 5%	10/20 50%
	Physician (*n* = 19)	4/20 20%	6/15 40%	2/20 10%	7/20 35%
	Psychologist/ Mental Health Specialist (*n* = 27)	4/20 20%	7/15 47%	11/20 55%	5/20 20%
	Not Mentioned (*n* = 24)	11/20 55%	4/15 27%	8/20 40%	1/20 5%
Treatment Details	Time to Completion Mentioned (*n* = 15)	3/20 15%	11/15 73%	0/20 0%	1/20 5%
	Treatment Costs Mentioned (*n* = 3)	0/20 0%	1/15 7%	0/20 0%	2/20 10%
	Financial Assistance for Treatment Mentioned (*n* = 13)	2/20 10%	11/15 73%	0/20 0%	0/20 0%
	Treatment Delivery Method Mentioned (*n* = 20)	3/20 15%	11/15 73%	2/20 10%	4/20 20%
Websites that Mentioned PTSD Programs (*n* = 29)		4/20 20%	14/15 93%	6/20 30%	5/20 25%
Websites that Mentioned Self-Help Advice (*n* = 44)		15/20 75%	6/15 40%	16/20 80%	7/20 35%
PTSD Information Provided	PTSD/OSI Definitions (*n* = 23)	2/20 10%	11/15 73%	7/20 35%	3/20 15%
	Risk Factors and Causes of PTSD (*n* = 34)	13/20 60%	6/15 40%	16/20 80%	16/20 80%
	PTSD Symptoms (*n* = 49)	10/20 50%	6/15 40%	20/20 100%	13/20 65%
	PTSD Diagnosis (*n* = 26)	6/20 30%	3/15 20%	8/20 40%	9/20 45%
	PTSD Statistics (*n* = 9)	8/20 35%	0/15 0%	0/20 0%	1/20 5%
	Impacts on Daily Living (*n* = 10)	3/20 15%	2/15 13%	1/20 5%	4/20 20%
	Nothing Mentioned (*n* = 5)	1/20 5%	1/15 7%	0/20 0%	3/20 15%

**Table 3 ijerph-17-07629-t003:** The 10 best websites based on the total DISCERN score and rating category, the Flesch–Kincaid Grade, and the number of treatments the websites mentioned that have been supported by strong evidence. The average DISCERN total scores are out of a total possible score of 75, with higher scores representing better quality websites. The average Flesch–Kincaid Grade levels represent school grade levels, with lower grades being simpler to read and understand. There is a total of three treatments supported by strong evidence (i.e., exposure therapy, CBT, and EMDR). The category ‘First Responder Focused’ refers to whether or not the website is targeted towards first responders (including firefighters).

Website	Total DISCERN Score	Flesch Kincaid Grade	Number of Treatments Mentioned Supported by Strong Evidence	First Responder Focused
National Institute of Mental Health (NIMH)—Post-Traumatic Stress Disorder [54]	66 (excellent)	9.8	2 (Exposure therapy, CBT)	No
Deciding to Get Treatment for PTSD| Healthlink British Columbia (BC) [55]	65 (excellent)	5.4	3 (Exposure therapy, CBT, EMDR)	No
Phoenix Australia [56]	64 (excellent)	9.4	2 (CBT and EMDR)	Yes
PTSD Treatment|Veterans Affairs [57]	63 (excellent)	6.7	3 (Exposure therapy, CBT, EMDR)	No
PTSD Treatment Basics—PTSD: National Center for PTSD [58]	62 (excellent)	8.1	3 (Exposure therapy, CBT, EMDR)	No
Everyday Health [59]	61 (excellent)	8.6	3 (Exposure therapy, CBT, EMDR)	No
Veterans Affairs Canada [60]	60 (good)	6.6	3 (Exposure therapy, CBT, EMDR)	No
Wake Forest University (WFU) Online Counselling [61]	59 (good)	8.0	3 (Exposure therapy, CBT, EMDR)	No
Mayo Clinic—Post-traumatic stress disorder (PTSD) [62]	56 (good)	6.3	3 (Exposure therapy, CBT, EMDR)	No
Fire Engineering [63]	51 (good)	5.9	3 (Exposure therapy, CBT, EMDR)	Yes

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
