# Peer review of "The Quality, Readability, Completeness, and Accuracy of PTSD Websites for Firefighters"

_ijerph, 2020, doi:10.3390/ijerph17207629_

Round 1

Reviewer 1 Report

The topic is important and the authors have used several tools to study the subject, good.

There are a few items to consider.

-ABSTRACT: please clarify DISCERN

-scores to represent ‘marginal’, ‘good’, “very good” and excellent websites should be mentioned.

-by "evidence for medications appears inconsistent in the literature". Should the literature refer to literature on the websites?

-INTRODUCTION: please provide some examples of stigma that the firefighters experience as stigma is a big barrier for their help-seeking

-some literature in dated. For example, internet use is referring to year 2012. Please check and update the literature refer the most recent one.

-MATERIALS AND METHODS: What is meant by A second rater evaluated the website quality using the DISCERN for "some of the websites" to check for rating bias. How many were checked and what were the results compared to the 1st rater? Please specify.

-what is the basis for A list of codes that was developed a priori to aid in the extraction of data from the websites? Who made them? Isn't there any literature or classification that can be referred to.

-Table 1 is overemphasized in the manuscript. Could it be explained within a text, as an example.

-RESULTS: I do not quite understand why tell that The keyword searches on Google returned the following numbers of results: 802,000 for 218 firefighter PTSD, 3,750,000 for firefighter operational stress, 62,200,000 for PTSD treatment, and 219 67,600,000 for PTSD symptoms - as the results was 75 websites. How did you extract websites from the results?

-I am curious to know about the classification of the websites (psychologist, veterans, general public, first responders, general wesite etc). What is it based on?

-I truly question the authors decision to reveal the names of the best and especially the name of the worst webpages. I suggest to remove them or tell the results of all the websites.

-I suggest to explain the contents of DISCERN in figure 1. Some of the criteria are difficult to understand?

-I suggest to include the maximum number of the scale in table 2 and 4. As the scales are different, it is hard to understand/recall wether the representing numbers are good/poor etc.

-where do the coding categories of table 3 come from? Basis for the categories are needed.

-Tables 3 and 4 are difficult to understand. Consider representing them in written text.

-DISCUSSION: I think it should be mentioned in limitations that the study only focused in English web-pages

Author Response

Thank you for your feedback, it has been very beneficial to improving my paper. I have copied and pasted your comments and will reply to them below in red.

-ABSTRACT: please clarify DISCERN

"a quality assessment tool (DISCERN)" was added to the abstract 

-scores to represent ‘marginal’, ‘good’, “very good” and excellent websites should be mentioned.

The representative category ranges was provided in the abstract

-by "evidence for medications appears inconsistent in the literature". Should the literature refer to literature on the websites?

This refers to the current scientific literature (i.e. systematic reviews). This was clarified in the abstract

-INTRODUCTION: please provide some examples of stigma that the firefighters experience as stigma is a big barrier for their help-seeking

Added to the introduction: "Due to the ‘macho mentality’ prevalent in the male-dominated occupation of firefighting, firefighters may be seen as weak if they report mental health issues [8]. Firefighters may be worried about their reputation and see mental illness as a threat to their career, thus they may be less likely to report PTSD or seek professional help [8]."

-some literature in dated. For example, internet use is referring to year 2012. Please check and update the literature refer the most recent one.

Unfortunately this is the most updated reference for these specific statistics. Canada is in the process of performing an updated one (no online health information has been released) but the last one was performed in 2012 and this is the same for the American data. Nothing appears to have been done specific to health information online recently in either Canada or the USA.

I was able to find the Canadian statistics published in 2013 which were also added in the introduction.

-MATERIALS AND METHODS: What is meant by A second rater evaluated the website quality using the DISCERN for "some of the websites" to check for rating bias. How many were checked and what were the results compared to the 1st rater? Please specify.

Added to the methods to clarify: "The first three randomly selected websites were evaluated by the primary rater and a second rater using the same DISCERN criteria to check for rating bias and to make sure the primary rater understood how to use the rating system. After performing the quality assessment for the three websites separately, the two evaluators met to discuss their ratings until a consensus was reached. There was agreement between the raters for 2 of the websites. One website was found to have different ratings as the first rater had reversed the rating scale. This error was discussed and corrected."

No formal testing of agreement was performed as the primary rater was responsible for data collection and the second rater was only responsible for overviewing the process to make sure the primary rater (a junior researcher) understood how to use the rating system.

-what is the basis for A list of codes that was developed a priori to aid in the extraction of data from the websites? Who made them? Isn't there any literature or classification that can be referred to.

The list of codes was original to our study and was developed by co-author consensus.

Added in the methods to explain: "An original list of codes specific to PTSD online content for firefighters was developed a priori to aid in the extraction of data from the websites. The codes were generated by two co-authors with research experience in the field of mental health and PTSD in public safety personnel, and then these codes were revised by the entire research team. The final list of codes was developed by co-author consensus."

-Table 1 is overemphasized in the manuscript. Could it be explained within a text, as an example.

Agreed the table was removed – details were added in the methods section “The Development of Evidence Based Statements”

-RESULTS: I do not quite understand why tell that The keyword searches on Google returned the following numbers of results: 802,000 for 218 firefighter PTSD, 3,750,000 for firefighter operational stress, 62,200,000 for PTSD treatment, and 219 67,600,000 for PTSD symptoms - as the results was 75 websites. How did you extract websites from the results?

Agreed, the information about the total number of Google results was removed as only the first 20 relevant websites were used for the data analysis in order to make this search as relevant and realistic to a real search a firefighter would do

 In the methods section:  "Most people searching for information online will stop at the second page of Google results, meaning approximately 20 results are typically reviewed [28]. The first 20 results that fit the inclusion criteria for each keyword search were included in this study, unless 20 relevant results could not be found. If less than 20 results were found for a search term, all relevant results were included. The name of the websites and the website hyperlinks were recorded in a spreadsheet organized by search term."

-I am curious to know about the classification of the websites (psychologist, veterans, general public, first responders, general wesite etc). What is it based on?

This was based on the discretion of the researcher who performed the content analysis based on guidelines:

- Was a specific population mentioned (if no this is just a general population; if yes who specifically?)

- We also were specifically interest in whether or not the website was specific to or targeted towards firefighters or first responders 

- Typically websites would specific say who the target of the information or the program was – the terms used on the website were used to describe the target population in the study

  • From our codebook (supplementary document attached to this submission), this was the goal of the target audience code: Who is the target audience for this program/website? For example, firefighters? Other emergency personnel like police? Do they speak to volunteer or career firefighters, or both?

-I truly question the authors decision to reveal the names of the best and especially the name of the worst webpages. I suggest to remove them or tell the results of all the websites.

Agreed regarding the worst websites – this information was removed throughout the paper.

The only information specific to websites was kept for the top 20 list of best websites as this was done in order to provide a resource specifically for firefighters as part of our FIREWELL.ca research association

-I suggest to explain the contents of DISCERN in figure 1. Some of the criteria are difficult to understand?

The criteria descriptions were added to Figure 1 

-I suggest to include the maximum number of the scale in table 2 and 4. As the scales are different, it is hard to understand/recall wether the representing numbers are good/poor etc.

Scaling details were included in figure 2 and 4

-where do the coding categories of table 3 come from? Basis for the categories are needed.

The coding sheet outlining this are provided in the supplementary documents

A the codes and a brief description were provided in the methods: "The codes and the guiding definitions used in the data extraction included: recommended PTSD treatment and program details (e.g. type of program or treatment, delivery methods, time to completion, associated costs, evidence of effectiveness, associated pseudoscience), treatment providers (e.g. clinical psychologist, certified trainer, peer), website’s target audience (e.g. general information seekers, specific populations like firefighters, or people with PTSD looking for support programs), self-help advice provided, and information about PTSD (e.g. causes, diagnosis, symptoms, quality of life)."

-Tables 3 and 4 are difficult to understand. Consider representing them in written text.

We do believe that the tables are the most succinct way to summarize these results in an organized manner. The tables have been spaced out better to make them easier to read. The titles are now more descriptive to make the interpretations simpler.

-DISCUSSION: I think it should be mentioned in limitations that the study only focused in English web-pages

Agreed - this was mentioned as a limitation in the discussion section

Reviewer 2 Report

This study includes a content analysis of websites providing information regarding PTSD and treatments, with a focus on those aimed at firefighters in Canada. Strengths of the study include its high degree of applied relevance and use of multiple well-validated measures of communication quality. I have a few comments that I think should be addressed. 1. The influence of Google search localization settings on the sample of websites produced by the search. The results state that many of the search results focused on veterans of the Canadian military and the RCMP -- since the search criteria did not specify “Canada” or related terms, the fact that these were produced is indicative of the likelihood that the same search is likely to yield a different sample of websites when conducted in different countries. The quality of health communication might also differ to some extent between countries, since websites originating in different countries might also differ in terms of their target audiences, culture, and local healthcare infrastructure. These issues should be addressed in the methods (e.g., was the search conducted using the google.ca domain?) and the discussion sections. 2. Given the specific national context of the study, as discussed above, this information should be included in the abstract and possibly also in the title (although I think it is probably not absolutely necessary in the title). 3. Figure 2 is difficult to read and interpret. Consider either incorporating this information in a table, or finding a way of combining DISCERN categories to summarize the results more succinctly.

Author Response

Thank you very much for your feedback, it has been very helpful in improving the paper. I have copied and pasted your comments and I will reply to them in red.

This study includes a content analysis of websites providing information regarding PTSD and treatments, with a focus on those aimed at firefighters in Canada. Strengths of the study include its high degree of applied relevance and use of multiple well-validated measures of communication quality. I have a few comments that I think should be addressed. 1. The influence of Google search localization settings on the sample of websites produced by the search. The results state that many of the search results focused on veterans of the Canadian military and the RCMP -- since the search criteria did not specify “Canada” or related terms, the fact that these were produced is indicative of the likelihood that the same search is likely to yield a different sample of websites when conducted in different countries. The quality of health communication might also differ to some extent between countries, since websites originating in different countries might also differ in terms of their target audiences, culture, and local healthcare infrastructure. These issues should be addressed in the methods (e.g., was the search conducted using the google.ca domain?) and the discussion sections. 2. Given the specific national context of the study, as discussed above, this information should be included in the abstract and possibly also in the title (although I think it is probably not absolutely necessary in the title).

I agree – Google.ca (Canada) was added in the abstract, methods and was discussed in the discussion/limitation. I did not add this to the title though because not all websites were Canadian and is still relevant to firefighters outside of Canada

For example, in the limitations I added: "The results might have also been different if a different Google location was used (i.e. Google.com). Because the search was performed in Canada, Google.ca was used and thus, Canadian websites were primarily identified. It is likely that if this search was performed in another country that other websites with different content and target audiences would be found. Because of this, the results of the content analysis are most likely specific to a Canadian Google search, and thus relevant only to Canadian firefighters. Although both English and French are the official languages of Canada, only English websites were searched. It is a limitation that French websites and websites of other languages were not included in the study. Moreover, the keywords selected had a large impact on the websites that were found in the current study. For example, the term ‘operational stress’ is often used for public safety personnel (i.e. firefighters, paramedics, police officers, RCMP) as well as for military personnel and veterans (i.e. past or present members of the Canadian Armed Forces) [7], and thus this influenced the results of the content analysis. "

3. Figure 2 is difficult to read and interpret. Consider either incorporating this information in a table, or finding a way of combining DISCERN categories to summarize the results more succinctly. 

DISCERN scores have been combined for clarity and a brief summary of these results are provided in text. A description of each DISCERN criteria is provided to help clarify the results provided in figure 1 and 2